# Phosphineoxide-Chelated Europium(III) Nanoparticles for Ceftriaxone Detection

**DOI:** 10.3390/nano13030438

**Published:** 2023-01-21

**Authors:** Rustem Zairov, Alexey Dovzhenko, Natalia Terekhova, Timur Kornev, Ying Zhou, Zeai Huang, Dmitry Tatarinov, Guliya Nizameeva, Robert R. Fayzullin, Aidar T. Gubaidullin, Taliya Salikhova, Francesco Enrichi, Vladimir F. Mironov, Asiya Mustafina

**Affiliations:** 1Arbuzov Institute of Organic and Physical Chemistry, FRC Kazan Scientific Center, Russian Academy of Sciences, 8 Arbuzov Str., 420088 Kazan, Russia; 2Institute of Chemistry, Kazan Federal University, Kremlevskaya Str. 18, 420008 Kazan, Russia; 3Institute of Carbon Neutrality, School of New 401 Energy and Materials, Southwest Petroleum University, Chengdu 610500, China; 4Department of Nanomaterials and Nanotechnology, Kazan National Research Technological University, 68, Karl Marx Str., 420015 Kazan, Russia; 5Institute of Fundamental Medicine and Biology, Kazan (Volga Region) Federal University, 18 Kremlevskaya Str., 420008 Kazan, Russia; 6Department of Computer Science, University of Verona, Strada Le Grazie 15, 37134 Verona, Italy; 7CNR-ISP, Institute of Polar Science of the National Research Council, Via Torino 155, 30174 Venezia, Italy

**Keywords:** luminescent sensor, lanthanide-centered luminescence, bis(phosphine oxide), cephalosporins, Judd-Ofelt

## Abstract

The present work demonstrates the optimization of the ligand structure in the series of bis(phosphine oxide) and β-ketophosphine oxide representatives for efficient coordination of Tb^3+^ and Eu^3+^ ions with the formation of the complexes exhibiting high Tb^3+^- and Eu^3+^-centered luminescence. The analysis of the stoichiometry and structure of the lanthanide complexes obtained using the XRD method reveals the great impact of the bridging group nature between two phosphine oxide moieties on the coordination mode of the ligands with Tb^3+^ and Eu^3+^ ions. The bridging imido-group facilitates the deprotonation of the imido- bis(phosphine oxide) ligand followed by the formation of tris-complexes. The spectral and PXRD analysis of the separated colloids indicates that the high stability of the tris-complexes provides their safe conversion into polystyrenesulfonate-stabilized colloids using the solvent exchange method. The red Eu^3+^-centered luminescence of the tris-complex exhibits the same specificity in the solutions and the colloids. The pronounced luminescent response on the antibiotic ceftriaxone allows for sensing the latter in aqueous solutions with an LOD value equal to 0.974 μM.

## 1. Introduction

Luminescent lanthanide complexes have long been attracting researchers due to their excellent optical properties. They are widely used as building blocks for luminescent nanomaterials for application in biomedical analysis, medical diagnosis, and cell imaging [1,2,3]. Long luminescence lifetimes, sharp characteristic emission bands, and large Stoke’s shifts allow the removal of biological background autofluorescence, which is of great importance when biomedical experiments are conducted.

However, due to the difficulties with direct sensitization of Ln(III), luminescence organic ligands (Ls) are usually used to excite lanthanides indirectly mainly with ligands’ triplet level [4,5,6,7]. Within the complex of Ln(III)–L, the efficiency of ligand-to-metal energy transfer governs the resulting brightness and, hence, luminescence performance in various applications. It is worth noting that β-diketones demonstrated outstanding chelating properties towards all lanthanide series forming stable six-membered chelate rings [8,9,10]. The structural variability of beta-diketones gave rise to a number of Ln(III) complexes with record optical properties due to the precise tuning of triplet energy level with a simple change of substituents [11,12,13]. Moreover, a great number of such complexes were successfully converted into aqueous colloids using the solvent exchange technique [8,14]. However, 1,3-diketonates are quite selective in sensitizing specific lanthanide ions. In this regard, the design of new ligands that can excite different metals of the lanthanide series and nanomaterials exhibiting both diverse lanthanide-centered luminescence in the visible region of the spectrum, e.g., green or red, and good colloid properties simultaneously is a top challenge in current chemistry.

Phosphine oxides attracted great attention as promising extractants for 4(5)f-metal ions [15,16,17,18,19,20]. The chemistry of 1,3-bis-phosphine oxides is rapidly developing, which is mainly because bis(phosphine oxides) are efficient ligands allowing diverse complexation with lanthanide and calcium ions [21,22,23,24,25,26]. The great impact of the structure of the group linking two phosphine oxides on the complexation mode is worth noting. In particular, the chelation of lanthanide ions by imido-bis(phosphine oxides) is followed by a deprotonation [27,28,29], while bis(phosphine oxides) bridged via methylene group tend to form stable complexes with lanthanide ions without deprotonation [30,31]. A similar chelating ability towards lanthanide ions was previously demonstrated for carbamoylphosphine oxide derivatives [32,33,34]. Having a valence of five, the phosphorus atom brings an additional option for introducing P-substituent compared to dicarbonyles. This makes carbamoylphosphine oxides (CMPOs) and bis(phosphine oxides) beneficial in terms of structural variations and possible functional add-ons. Aryl-substituted phosphorus-containing ligands have recently been documented to grant substantial antenna-effect sensitizing luminescence of Rare Earth Elements (REEs), which are in demand in luminescence-related applications [35,36]. However, to the best of our knowledge, the applicability of such complexes in the development of nanosensors is very poorly documented in the literature, if at all.

The wide applicability of lanthanide complexes in the fluorescent sensing of residual amounts of drugs, including antibiotics, in water or biological fluids is well known [37,38,39,40,41,42,43]. In this regard, the synthesis of new lanthanide complexes, where ligand-to-metal energy transfer is enough for sensitizing lanthanide-centered luminescence, and where ligand-metal coordination bonds are tight enough for the safe conversion of the complexes into water-dispersible nanomaterial with high lanthanide-centered luminescence, is a challenging scientific task.

Thus, the present work represents the already known [29,30] (Figure 1a,b) and newly synthesized (Figure 1c,d) ligands for Tb^3+^ and Eu^3+^ ions. An additional methoxyphenyl fragment was purposely introduced at the alpha-position to promote further build-ups, to control the lanthanide coordination center symmetry for additional boosting lanthanide-centered luminescence, and for additional hydrophobic shielding of the inner-sphere of the lanthanide ions minimizing solvent-induced nonradiative relaxation. The synthesis of the complexes and their structural and spectral characterization in the solid state and solutions are described in detail in this work. The efficiency of converting the as-synthesized complexes into the aqueous PSS-stabilized colloids is correlated with the structures of the complexes and the ligands. The diverse physico-chemical techniques applied for the characterization of the ligands, complexes, and nanoparticles on molecular, supramolecular, and nano-levels are focused on the ability of the ligands to sensitize the lanthanide-centered luminescence in both organic solutions and in the aqueous colloids. The applicability of the developed aqueous colloids as nanosensors will be demonstrated using their luminescent reply on ceftriaxone, which is the third generation of the cephalosporin antibiotic widely applied in treating such socially relevant bacterial infections as meningitis, pneumonia, and many others [44,45,46,47].

## 2. Experimental Section

Materials. All reagents were used as purchased from Sigma-Aldrich or Acros Chemicals without further purification. Solvents were purified using standard procedures before use. All reactions were run under an argon atmosphere unless in aqueous media. Ceftriaxone disodium salt from RUE Belmedpreparaty was used as purchased.

Syntheses. Ligand **a** was obtained using a procedure suggested by Magennis et al. [28], and ligand **b** was synthesized using a method described by Maass et al. [48]. Detailed synthetic procedures as well as their lanthanide complexes syntheses, [Ln(**a**)_3_] and [Ln(**b**)_2_(NO_3_)_3_], are given in SI. Synthetic protocols and characterization using ^1^H NMR, IR spectra, ^31^P NMR, and ESI-MS for ligands **c** and **d** are described in detail in ESI.

Nanoparticle synthesis. For each system, 0.5 mL of a solution of the corresponding complex (C = 3 mM) in acetonitrile was added dropwise with a syringe plunger into 2.5 mL of PSS (sodium polystyrene sulfonate) aqueous solution (1 g/L) containing NaCl (C = 0.5 M) while vigorously stirring (2200 rpm). The formation of fine white colloids was observed. To separate the colloid nanoparticles from acetonitrile, the mixtures were centrifuged, and the supernatant was removed and replaced with H_2_O (V = 3 mL). These operations were repeated twice. After each replacement of the solvent, the solutions were ultrasonicated for 10 min.

Methods. Mass spectra were recorded with an AmaZon X «Bruker» mass spectrometer. IR spectra were recorded with a Bruker Tensor-27 instrument for the samples in KBr pellets. NMR experiments were carried out with 400 MHz [400 MHz (^1^H), 162 MHz (^31^P)] or 600 MHz [600 MHz (^1^H), 243 MHz (^31^P),] spectrometers equipped with a pulsed gradient unit capable of producing magnetic field pulse gradients in the z-direction of 53.5 G cm^–1^. All spectra were acquired in a 5 mm gradient inverse broadband probe head. Chemical shifts (d) are expressed in parts per million, relative to the residual ^1^H signal of CDCl_3_, and the signals are designated as follows: s, singlet; d, doublet; t, triplet; m, multiplet. Coupling constants (J) are in hertz (Hz). Excitation and emission spectra and luminescence decay curves were registered with Fluorolog-QM (Horiba). The transmission electron microscopy (TEM) was performed using a Hitachi HT7700 (tungsten filament, HV = 100 kV). Samples were deposited on a 300-mesh copper grid with continuous carbon-formvar support film. XRD of crystals was performed with Bruker D8 QUEST. Detailed descriptions can be found in Electronic Appendix A. Powder X-ray diffraction (PXRD) measurements were performed with a Bruker D8 Advance diffractometer equipped with a Vario attachment and Vantec linear PSD, using Cu radiation (40 kV, 40 mA) monochromated with the curved Johansson monochromator (λ Cu Kα1 1.5406 Å). Electron absorption spectra were registered with Analytic Jena SPECORD 50plus, for dynamic light scattering and zeta-potential measurements Malvern Zetasizer Nano ZS was utilized. The fundamental basis of Judd–Ofelt analysis [49,50] and corresponding formulas [51,52] are collected in ESI.

## 3. Results and Discussion

### 3.1. Synthesis of Ligands **c** and **d**

Ligands **a** and **b** were obtained and reported previously [28,48]. Both syntheses were reproduced in this work in order to synthesize a row of lanthanide(III) complexes to conduct a comparative study. Ligands **c** and **d** were obtained using known methods with some adjustments as depicted in Figure 1. In brief, ligand **c** was synthesized via the interaction of 1-(dichloromethyl)-4-methoxybenzene and ethyl diphenylphosphinite, including Arbuzov rearrangement at high temperature neat until the reaction mixture solidified. The desired ((4-methoxyphenyl)methylene)bis(diphenylphosphine oxide) **c** was obtained individually with recrystallization as a white poorly-soluble solid. Ligand **d** was synthesized via the oxidation of a crude reaction mixture of corresponding phosphine obtained as described by Ogawa et al. [53]. The desired *N*-(diphenylphosphoryl)-*N*-(4-methoxyphenyl)-*P*,*P*-diphenylphosphinic amide **c** was obtained individually with two subsequent recrystallizations.

### 3.2. Synthesis of [Ln(L)x] Complexes, Ln = Eu^3+^, Tb^3+^, L = a, b, c, x = 1,2,3 and XRD Data

Lanthanide complexes of bis(phosphine oxide) (**a**) and 2-diphenylphosphineoxide-1-phenylethanone (**b**) were obtained previously and reproduced in this work in order to compare luminescent properties within the row of complexes [29,30]. X-ray crystal structures of [Tb(**a**)_3_] and [Eu(**b**)_2_(NO_3_)_3_] are illustrated in Figure 2a and 2b, respectively.

Single crystals of complex [Tb(**c**)(NO_3_)_3_(H_2_O)] as colorless prisms were prepared using slow evaporation of a THF solution. The compound crystallizes in the monoclinic space group *C*2/*c* as a crystallosolvate with 2.5 THF molecules per complex. One complex molecule is present in the asymmetric cell. Its molecular structure is shown in Figure 2c,d and reflects the 1:1 stoichiometry. Nine-coordinated Tb atom is surrounded by one ligand **c** attached by bis(phosphine oxide) chelation through the P=O coordination, three ditopic NO_3_^-^ residues, and one water molecule. Interestingly, the [P=]O–Tb internuclear distances are notably different and equal to 2.368(1) and 2.295(1) Å. The internuclear distances between the Tb and coordinated oxygen atoms of NO_3_^-^ vary between 2.430(2) and 2.507(1) Å. Deposition number CCDC 2,218,909 contains the Appendix A for compound [Tb(**c**)(NO_3_)_3_(H_2_O)]. These data are provided free of charge by the joint Cambridge Crystallographic Data Centre and Fach informations zentrum Karlsruhe Access Structures service www.ccdc.cam.ac.uk/structures (Accessed on 10 November 2022).

X-ray structures of previously studied [Tb(**a**)_3_] and [Eu(**b**)_2_(NO_3_)_3_], as well as the newly obtained [Tb(**c**)(NO_3_)_3_(H_2_O)], are presented in Figure 2a–c to illustrate the difference in the inner-sphere environment of the lanthanide ions in the complexes. In particular, the unsaturated coordination environment of Tb^3+^ ion in [Tb(**a**)_3_] by six oxygen atoms derives from the chelation of three imido-bis-phosphinates. The coordination polyhedron of [Eu(**b**)_2_(NO_3_)_3_] is filled by ten oxygen atoms arising from the coordination of both ligands and nitrates, and the nine oxygen atoms in the polyhedra of complex [Tb(**c**)(NO_3_)_3_(H_2_O)] also arises from the coordination of one organic ligand, three nitrates, and one water molecule. Complex [Tb(**a**)_3_] crystallizes with two molecules in the asymmetric cell. Although both molecules are situated at the crystallographic 3-fold proper rotating axes, they show different coordination modes, namely, trigonal prismatic and octahedral. Complexes [Eu(**b**)_2_(NO_3_)_3_] and [Tb(**c**)(NO_3_)_3_(H_2_O)] are located in general position in the crystals and, consequently, they are asymmetric.

Thus, the structure of ligands is of great impact on the complex structure. It is worth noting the conversion of the ligand from phosphine oxide to phosphonate forms as the reason for the formation of complex [Tb(**a**)_3_] with the coordinative unsaturated inner-sphere environment of the Tb^3+^ ion. A similar structure is revealed for some 1,3-diketonate complexes of lanthanides [54,55]. In turn, the substitution of N–H by N–R in the structure of **d** (Figure 1) results in poor complexation. One can hypothesize that an impossibility of the deprotonation of ligand **d** explains its poor ability to form complex, while the ability of the phosphine oxide representative **a** with N–H bridging moiety to deprotonate prerequisites its high complexing ability. The ligands **b** and **c** tend to chelate without deprotonation, which agrees well with the literature data [30,31]. The smaller electron donating properties of both bis(phosphine oxide) **c** and ketophosphine oxide **b** vs. the anionic deprotonated form of ligand **a** result in the formation of the saturated coordination sphere due to the coordination of the nitrates and water molecule along with the organic ligands. The aforesaid difference in the coordination polyhedra can be of some impact on the lanthanide-centered luminescence of the complexes. It is also worth noting that the difference in the complex stoichiometry, which is 1:2 and 1:1 (metal-to-ligand ratio) for the ligands **b** and **c**, respectively, can be explained by the additional steric constraints arising from the introduction of the bulky methoxyphenyl moiety in the ligand **c**.

### 3.3. Absorption and Luminescent Properties of Ln(**c**)

Absorption spectra of the newly synthesized ligands **c** and **d** as well as their terbium complexes were recorded in CH_3_CN (Figure 3a). Both ligands are characterized by an absorption band at 230 nm responsible for the π→π* transition. The spectrum of [Tb(**c**)(NO_3_)_3_(H_2_O)] differs significantly from that of the ligand.

The as-synthesized terbium and europium complexes with ligands **a**, **b**, and **c** are well soluble in CH_3_CN. Their luminescence spectra recorded at the same instrumental conditions revealed that terbium centers are more effectively sensitized by ligands **a** and **b** vs. **c** (Figure 3b). It is worth assuming that the tris-chelation of ligand **a** can be a reason for better sheltering of a coordination center from solvent molecules, while the inner-sphere nitrates in [Tb(**b**)_2_(NO_3_)_3_] and [Tb(**c**)(NO_3_)_3_(H_2_O)] can easily be substituted by the solvent molecules. However, the excited state lifetime (τ_meas_) values calculated from the luminescence decay kinetic measurements (Table 1) remain high, arguing against dissociation of the complexes [Tb(**b**)_2_(NO_3_)_3_] and [Tb(**c**)(NO_3_)_3_(H_2_O)].

The sensitivity to changes in symmetry and strength of the ligand field is the peculiar feature of the Eu^3+^-centered luminescence. The intensity ratio of ^5^D_0_→^7^F_2_/^5^D_0_→^7^F_1_ transitions (R) is the well-known value that allows for following the changes in the symmetry and strength of the ligand field around Eu^3+^ ions. This derives from the fact that the intensity of the dipole transition ^5^D_0_→^7^F_2_ is hypersensitive to electric symmetry and strength of the ligand field around Eu^3+^ ions, while the magnetic dipole transition ^5^D_0_→^7^F_1_ is not affected by the surrounding charge distribution because it is parity-allowed, and its emission is often used as internal standard [56]. Thus, the R-values calculated using Equation (1) are collected in Table 1 along with the τ-values calculated from the luminescence decay kinetic measurements.
(1)R=I(D50→F72)I(D50→F71)

The higher the R, the lower the local symmetry around Eu^3+^ is with respect to an inversion center since a high local symmetry strongly reduces the electric dipolar emission without affecting the magnetic dipolar one, and vice versa. In addition, in many complexes, the ratio is high due to an increase in the covalency of Eu–ligand coordinative bonds, which lowers the symmetry.

Focusing our attention on Eu complexes, the PL emission spectra show a much different shape for the three analyzed complexes, as an indication of the different environment surrounding the rare earth ion. Indeed, the R-value is the highest for complex [Eu(**a**)_3_] as an indication of a low symmetry surrounding the rare earth ion. The opposite is observed for complexes [Eu(**b**)_2_(NO_3_)_3_] and [Eu(**c**)(NO_3_)_3_(H_2_O)], where the R-values barely exceed 1.0, thus, indicating that the magnetic transition intensity is similar to the electric dipolar one.

The luminescence decay kinetics of the europium complexes also demonstrate the mono-exponential decay, which argues for a rather poor dissociation of the complexes (Figure 4). The lifetime results (Table 1) are in agreement with the previous considerations. Due to the forbidden nature of the internal 4f transitions, the probability of electric dipole recombination is low, resulting in long lifetimes in the millisecond range as it occurs for [Eu(**b**)_2_(NO_3_)_3_] and [Eu(**c**)(NO_3_)_3_(H_2_O)] complexes. However, when the surrounding environment introduces an asymmetric perturbation to the wave functions, this increases the dipole transition rate and the electric dipole emission component, while it reduces the radiative lifetime. Thus, the calculated radiative lifetime reported in Table 1 is shorter for the complex [Eu(**a**)_3_] and longer for the other two. The measured lifetimes follow the same trend.

### 3.4. Synthesis of PSS-[Ln(L)x] Nanoparticles, L = a, b, c, x = 1,2, 3, Ln = Tb, Eu

Chasing applicability for biomedical purposes, the complexes were converted to water colloids according to the solvent change method [8], where the precipitated species are incorporated into the PSS capsules through an electrostatic attraction. Thus, the charge of the precipitated species is of great impact on the efficacy of the incorporation. The surface charge of the colloidal species can be measured by measuring their electrokinetic potential (ζ) values. Dropwise addition of the complexes in the acetonitrile solution (C = 1 mM) to water results in the formation of the colloids, which were characterized by measurements of their size and ζ-values (Table 2). These values were measured for the colloids formed from the Eu^3+^ complexes with the ligands (Table 2). The positive ζ-values of the colloids formed from complexes [Eu(**b**)_2_(NO_3_)_3_] and [Eu(**c**)(NO_3_)_3_(H_2_O)] are rather anticipated, since the surface exposed complex units can undergo partial degradation, most probably, due to the release of nitrate anions. The significant negative surface charging of the colloids from complex Eu(**a**)_3_ also argues for its partial degradation, while the significant basicity of the released deprotonated ligand triggers a formation of hydroxyls under the interaction with water molecules.

It is worth noting that the nanoprecipitation of the complexes in the solutions of PSS and NaCl results in the formation of the aqueous colloids, which are manifested by the average size values less than those of the nanoprecipitates formed in the aqueous solutions. The PSS-stabilized colloids exhibit high colloid stability, while their uncoated counterparts aggregate and precipitate within several hours. The efficient incorporation of the nanoprecipitates into the PSS-based aggregates is rather anticipated for the lanthanide complexes with ligands **b** and **c** since their nanoprecipitates are positively charged, while the incorporation of the negatively charged nanoprecipitates of complex [Eu(**a**)_3_] into the aggregates is rather unusual. However, we can keep in mind the high concentration of the counterions derived from the high concentration level of NaCl in the PSS solutions, which can facilitate the incorporation of the [Eu(**a**)_3_]-based nanoprecipitates into the PSS-capsules.

The efficacy of the complex transformation from the acetonitrile solutions into the PSS-stabilized colloids can be assessed by the partial leaching of the lanthanide ions. Thus, the equilibrium concentrations of the lanthanide ions were measured in the aqueous colloids and two first supernatants after the phase separation and the washing procedure (for more details see the experimental section). The percentage of the conversion calculated from the data is represented in Table 3 for the complexes [Eu(L)_x_]. The most efficient conversion is observed for [Eu(**a**)_3_], which indicates poor leaching of Eu^3+^ ions from the nanoprecipitates. The leaching is more pronounced for complexes [Eu(**b**)_2_(NO_3_)_3_] and [Eu(**c**)(NO_3_)_3_(H_2_O)], which correlates with the less tight binding of Eu^3+^ ions by ligands **b** and **c** than by ligand **a**. It is worth mentioning that ligands **b** and **c** tend to coordinate without deprotonation, while the tightest binding of Eu^3+^ by ligand **a** is due to its deprotonation, which enforces the coordination bonds in [Eu(**a**)_3_].

Figure 5a illustrates the morphology of PSS-[Eu(**a**)_3_] colloids dried on the formvar coated substrate. There are 35–70 nm particles of PSS stuffed by multiple 2–8 nm sized cores built of [Eu(**a**)_3_] complexes. Such plum-duff architecture is the key to more efficient sensing compared to PSS-PSS-[Eu(**b**)_2_(NO_3_)_3_] and [Eu(**c**)(NO_3_)_3_(H_2_O)] colloidal species (Appendix A) due to a highly developed surface. The produced colloids exhibit Eu^3+^-centered luminescence (Figure 5b). The R-values of the colloids follow a similar tendency as for the complexes in solutions (Table 1). Thus, the specific R-value of [Eu(**a**)_3_] in the acetonitrile solution and in the PSS-stabilized colloids argues for the similarity in the inner-sphere environment of the Eu^3+^ ion in both molecular and colloidal forms. The excited state lifetime values of the colloids (Table 1) also reveal the specificity of [Eu(**a**)_3_] similar to the complexes in the solutions. The aforesaid agrees well with the safe conversion of [Eu(**a**)_3_] complexes into colloids (Table 3).

The dried PSS-[Eu(**a**)_3_] colloids were analyzed using the PXRD method (Figure 5c,d). The PXRD pattern of the colloids reveals their crystalline nature with very poor if any contribution of an amorphous phase (Figure 5c,d). The PXRD pattern simulated out of single crystal data of [Eu(**a**)_3_] is also shown in Figure 5c for comparison with that of PSS-[Eu(**a**)_3_] colloids in order to prove their composition. The vast majority of reflections in diffractograms of [Eu(**a**)_3_] and PSS-[Eu(**a**)_3_] coincide. At the same time, comparative analysis of the difractograms of the powder samples of PSS-[Eu(**a**)_3_] colloids and ligand **a** represented in Figure 5d reveals no coincidence between the PXRD patterns of ligand **a** and PSS-[Eu(**a**)_3_]. This allows us to conclude that the nanoparticles contain a pure crystalline phase of the complex without an admixture of the separately precipitated ligand phase.

### 3.5. Detection of Ceftriaxone Using PSS-[Eu(a)_3_] Colloids

The high chemical stability of [Eu(**a**)_3_] and relative hydrophobicity ensured the efficient transformation of [Eu(**a**)_3_] into PSS-[Eu(**a**)_3_], which makes these colloids the most promising basis for reliable sensing of substrates. Short screening of antibiotics was performed for PSS-[Eu(**a**)_3_] water dispersions in terms of possible luminescent response. No quenching was observed for amoxicillin, while ciprofloxacin detection could be achieved only at high concentrations (Appendix A). In the meanwhile, the presence of micromolar amounts of ceftriaxone resulted in significant changes in emission intensity. The main europium band at 612 nm demonstrated quenching for the factor of 7.23 when 87.5 μM ceftriaxone was added (Figure 6a). Deviation of I_0_/I from the linear law in Stern–Volmer coordinates indicates a mixed, static, and dynamic mechanism of quenching, which is typical for dark complexes formation (Figure 6b) [57,58]. In turn, the dark complex formation observed for PSS-[Eu(**a**)_3_] colloids in the presence of ceftriaxone argues for the ligand exchange disturbing the ligand-to-metal energy transfer of [Eu(**a**)_3_]. The presence of the carboxylate and hydroxy group of 1,2,4-triazin-3-yl moiety in the structure of sodium ceftriaxone (Scheme S3, inset) prerequisites its efficiency in the ligand exchange. The limit of detection (LOD), calculated as 3σ/S (where σ is the standard deviation for blank experiments and S is the slope of the linear segment of luminescence intensity vs. the concentration of analyte), is equal to 0.974 μM, which is comparable to the previously reported values (Table 4). For comparative purposes, the luminescent response of PSS-[Eu(**b**)_2_(NO_3_)_3_] to ceftriaxone was also monitored (Appendix A). The latter colloids demonstrate a less sensitive response to the substrate, which can be explained by their mixed composition.

Commonly, a level of ceftriaxone is evaluated in blood serum or urine. For further use of the PSS-[Eu(**a**)_3_] nanoparticles for ceftriaxone detection in blood serum, the BSA, glutamic, and aspartic acids in 0.01 M phosphate buffer interfering effect was estimated. The luminescence response of PSS-[Eu(**a**)_3_] to ceftriaxone evaluated in the solutions modeling blood serum (Appendix A in ESI) indicates that the applied concentration levels of the protein and amino acids provide an insignificant effect on the sensitivity of PSS-[Eu(**a**)_3_] to ceftriaxone. This argues for the applicability of PSS-[Eu(**a**)_3_] for bioanalytical purposes.

## 4. Conclusions

Summarizing, the present work demonstrates the three representatives of bis(phosphine oxides) and β-ketophosphine oxide as ligands for Tb^3+^ and Eu^3+^ ions. The nature of the bridging group between phosphine oxide moieties has a great impact on the complex structure. The specificity of the imido-group as the bridging one between two phosphoryl moieties phosphine oxide moieties, manifested by the possibility of conversion of imido- bis(phosphine oxides) to their phosphonate form, is the reason for the formation of complexes [Ln(**a**)_3_] (Ln = Eu^3+^, Tb^3+^) with the coordinative unsaturated inner-sphere environment of the Ln^3+^ ion. The β-ketophosphine oxide and bis(phosphine oxide) ligands with methylene bridging groups are efficiently coordinated to Ln^3+^ ions without deprotonation, while the complex ability is poor for the ligand possessing the N–R bridging group. The coordination of the ligands with the Ln^3+^ ions results in the efficient sensitization of both Tb^3+^- and Eu^3+^-centered luminescence. The ligand environment in the [Ln(**a**)_3_] complexes provides the specific ligand field symmetry, which is manifested by the specific spectral pattern and excited state lifetime value of [Eu(**a**)_3_]. Moreover, the high stability of the [Ln(**a**)_3_] complexes provides their safe conversion into PSS-[Ln(**a**)_3_] colloids, while the lanthanide complexes with carbamoylphosphine oxide and bis(phosphine oxide) ligands suffer from the partial degradation under the synthesis of the PSS-stabilized colloids. The red luminescence of PSS-[Eu(**a**)_3_] fitting to the wavelengths range of the so-called biological window exhibit the pronounced luminescent response on the antibiotic ceftriaxone, which allows for sensing the latter in aqueous solutions with the LOD value equal to 0.974 μM.

## Figures and Tables

**Figure 1 nanomaterials-13-00438-f001:**
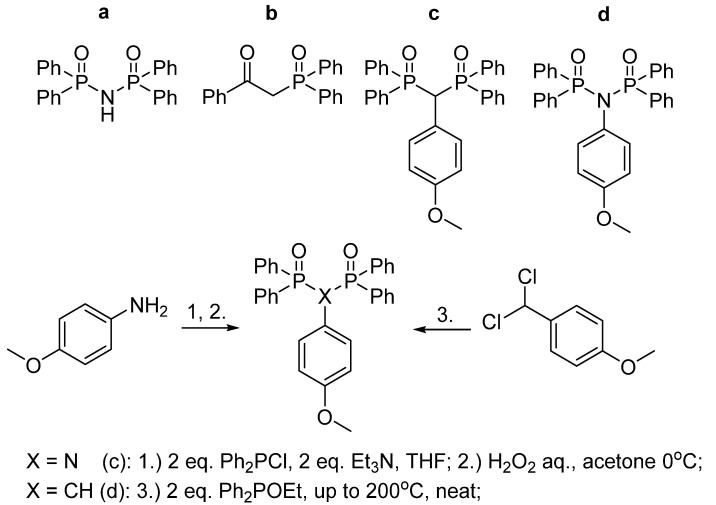
Ligands synthesized for this study: *N*-(diphenylphosphoryl)-*P*,*P*-diphenylphosphinicamide (**a**), 2-(diphenylphosphoryl)-1-phenylethan-1-one (**b**), ((4-methoxyphenyl)methylene)bis(diphenylphosphine oxide) (**c**), *N*-(diphenylphosphoryl)-*N*-(4-methoxyphenyl)-*P*,*P*-diphenylphosphinic amide (**d**). Synthetic routes employed in this work to synthesize new ligands **c** and **d** with *p*-methoxyphenyl substituent at *α*-position.

**Figure 2 nanomaterials-13-00438-f002:**
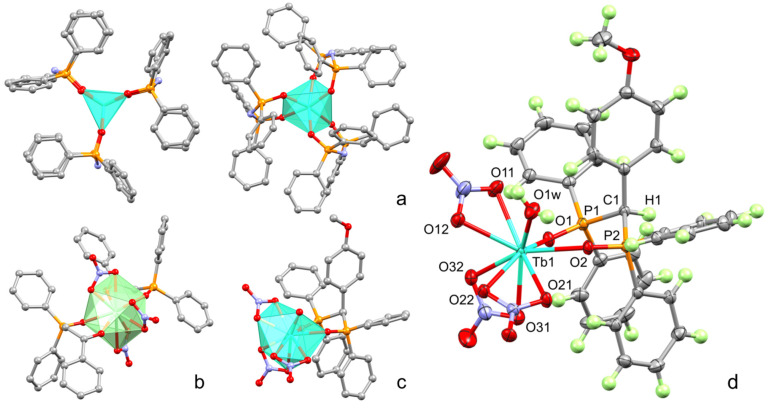
Comparison of the X-ray structures of complexes [Tb(**a**)_3_] (**a**), [Eu(**b**)_2_(NO_3_)_3_] (**b**), and [Tb(**c**)(NO_3_)_3_(H_2_O)] (**c**). Metal atoms are shown as polyhedra. Hydrogen atoms are omitted for clarity. In the case of [Tb(**a**)_3_], both molecules represented in the asymmetric cell are shown. ORTEP of complex [Tb(**c**)(NO_3_)_3_(H_2_O)] (**d**) at the 50% probability level for non-hydrogen atoms according to single-crystal X-ray diffraction. THF solvent molecules are omitted for clarity. Selected internuclear distances [Å]: Tb1–O1 2.368(1), Tb1–O2 2.295(1), Tb1–O1w 2.353(2), Tb1–O11 2.430(2), Tb1–O12 2.507(1), Tb1–O21 2.498(1), Tb1–O22 2.487(1), Tb1–O31 2.465(2), Tb1–O32 2.435(2), P1–O1 1.505(1), P2–O2 1.500(1).

**Figure 3 nanomaterials-13-00438-f003:**
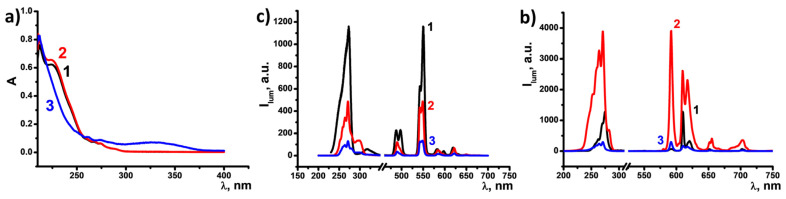
(**a**) UV-Vis absorption spectra of ligand **c** (1), **c** in the presence of triethylamine (2) and **c** in the presence of triethylamine and an equimolar amount of terbium(III) (C = 10^–5^ M). Excitation and emission spectra of [Ln(**a**)_3_] (1), [Ln(**b**)_2_(NO_3_)_3_] (2), [Ln(**c**)(NO_3_)_3_(H_2_O)] (3), (Ln = Tb^3+^ (**b**); Ln = Eu^3+^ (**c**)) (C_Ln_ = 10^–5^ M).

**Figure 4 nanomaterials-13-00438-f004:**
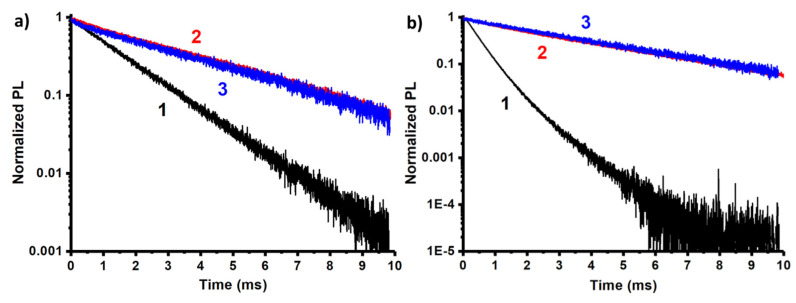
Luminescence decay curves for [Eu(**L**)_x_] (**a**) and PSS-[Eu(**L**)_x_], (**b**) in logarithmic scale (L = **a**(1), **b**(2), **c**(3)) under the excitation of corresponding wavelengths (Table 1).

**Figure 5 nanomaterials-13-00438-f005:**
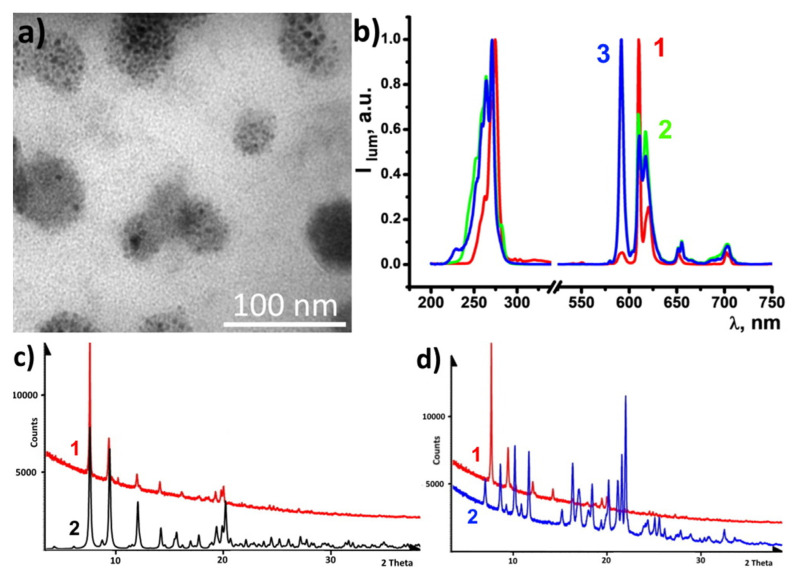
(**a**) TEM image of PSS-[Eu(**a**)_3_]. (**b**) Excitation and emission spectra of PSS-[Eu(**L**)_x_] (L = **a**, **b**, **c**, x = 3,2,1, respectively). (**c**) Simulated PXRD pattern out of single crystal data of [Eu(**a**)_3_] (2) in comparison to dried PSS-[Eu(**a**)_3_] PXRD pattern (1). (**d**) PXRD patterns of dried PSS-[Eu(**a**)_3_] (1) and ligand **a** (2).

**Figure 6 nanomaterials-13-00438-f006:**
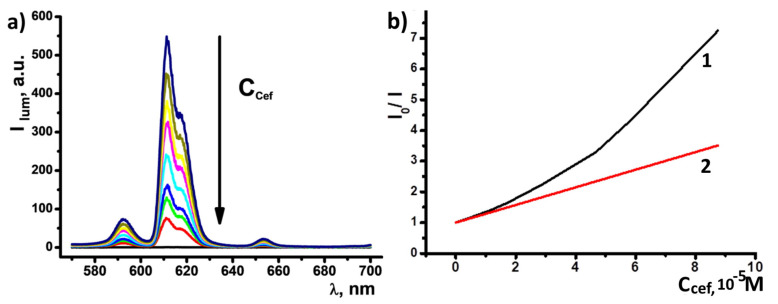
(**a**) Gradual quenching of Eu^3+^ luminescence within the composition of PSS-[Eu(**a**)_3_] colloids in the presence of increasing amounts of ceftriaxone (C = 57.4 μM). (**b**) I/I_0_ values vs. concentration of ceftriaxone, I_0_ and I are the luminescence intensities of PSS-[Eu(**a**)_3_] colloids without and in the presence of a certain concentration of ceftriaxone, respectively (1). Tangent for the curve (1) in the low-concentration region used for LOD calculation.

**Table 1 nanomaterials-13-00438-t001:** Decay parameters and excitation wavelengths (λ_exc_) for [Eu(**L**)_x_] in CH_3_CN.

	R	Tau_Rad (ms)	l_exc_ (nm)	Tau_Meas(ms)	PL Efficiency
[Tb(a)_3_]	-	-		2.20	
[Tb(b)_2_(NO_3_)_3_]	-	-		4.79	
[Tb(c)(NO_3_)_3_(H_2_O)]	-	-		3.05	
[Eu(a)_3_]	11.22	1.49	275	1.5	1.00
[Eu(b)_2_(NO_3_)_3_]	1.52	6.86	271	3.90	0.57
[Eu(c)(NO_3_)_3_(H_2_O)]	1.38	7.40	271	4.20	0.57
PSS-[Eu(a)_3_]	12.18	1.41	273	0.46	0.33
PSS-[Eu(b)_2_(NO_3_)_3_]	1.17	8.10	271	3.50	0.43
PSS[Eu(c)(NO_3_)_3_(H_2_O)]	0.86	10.13	271	3.33	0.33

**Table 2 nanomaterials-13-00438-t002:** DLS data (hydrodynamic diameter (d^h^), polydispersity index (PDI), and electrokinetic potential (ζ)) for [Eu(**L**)_x_] precipitates and PSS-[Ln(**L**)_x_] colloids in water.

Name	d^h^ (nm)	PDI	ζ (mV)
[Eu(a)_3_]	1305.0 ± 62.7	0.199 ± 0.182	–32.9 ± 0.6
[Eu(b)_2_(NO_3_)_3_]	878.6 ± 28.9	0.659 ± 0.016	35.7 ± 0.5
[Eu(c)(NO_3_)_3_(H_2_O)]	1922.0 ± 133.5	0.265 ± 0.157	28.3 ± 1.2
PSS-[Eu(a)_3_]	314.9 ± 11.2	0.272 ± 0.006	–27.1 ± 0.7
PSS-[Eu(b)_2_(NO_3_)_3_]	522.2 ± 58.3	0.549 ± 0.040	–67.1 ± 0.8
PSS-[Eu(c)(NO_3_)_3_(H_2_O)]	646.9 ± 68.5	0.561 ± 0.038	–62.0 ± 2.3

**Table 3 nanomaterials-13-00438-t003:** The equilibrium Eu^3+^ concentrations in the aqueous colloids and the supernatants after the phase separation and the washing procedure along with the percentage of Eu^3+^ in the colloids and supernatants.

	Introduced	Remain within the Composition of Colloids		C_Ln_ in Supernatant 1		C_Ln_ in Supernatant 2		Total Loss of Ln(III)	
	мM	мM	%	мM	%	мM	%	мM	%
PSS-[Eu(a)_3_]	0.5	0.344	68.94	0.13	26.50	0.023	4.55	0.155	31.05
PSS-[Eu(b)_2_(NO_3_)_3_]	0.5	0.108	21.69	0.37	73.49	0.024	4.82	0.392	78.31
PSS-[Eu(c)(NO_3_)_3_(H_2_O)]	0.5	0.116	23.29	0.38	37.67	0.007	1.38	38.36	76.71

**Table 4 nanomaterials-13-00438-t004:** LOD values of ceftriaxone detection for various luminescent nanomaterials/compounds reported in the literature.

Luminescent Compound	LOD (M)
CdSe/CdS/ZnS quantum dots [59]	1 × 10^–6^
Ceftriaxone converted into a fluorescent compound [60]	3.5 × 10^−8^
Chemiluminescence emission generated from the oxidation of ceftriaxone sodium [61]	4.5 × 10^–8^
Ceftriaxone converted into a fluorescent product [62]	2.3 × 10^–9^
Carbonized blue crab shell carbon dots [63]	9.0 × 10^–9^
Chicken drumstick-derived carbon dots [64]	4.4 ×10^–10^
Graphene quantum dots in a molecularly imprinted polymer MIP-GQDs [65]	1.8 × 10^–10^
L-cysteine (Cys) coated CdS QDs [66]	1.3 × 10^–9^
L-cysteine capped ZnS (L-Cys-ZnS) QDs [67]	9.0 × 10^−8^
Our paper	9.7 × 10^–7^

## Data Availability

The data presented in this study are available in the article and electronic Appendix A.

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
