# Peer review of "Phosphineoxide-Chelated Europium(III) Nanoparticles for Ceftriaxone Detection"

_nanomaterials, 2023, doi:10.3390/nano13030438_

Round 1

Reviewer 1 Report

The authors reported a kind of phosphine oxide-chelated EU3+-Nanoparticles for ceftriaxone detection, it shows a typical sharp-band f-f transitions from Eu3+ responding to the antibiotics ceftriaxone, I recommend it to be published on Nanomaterials after minor revision, some questions and comments are listed below:

1.      Fig. 3, in the absorption spectrum, what’s the reason of absence of f-f transitions of Eu3+, and in the PL spectrum, the fine structure of the nano-particles is clearly observed, which often the case for bulk materials with well crystallization while for nanoparticles, inhomogeneous broadening is often observed, please make some comments.

2.      Fig. 4, the lifetime of the sample is obviously over the detecting range of 10 ms, which equation did the author use for the fitting? Besides, the baseline should be carefully evaluated before lifetime fitting.

Author Response

The authors reported a kind of phosphine oxide-chelated EU3+-Nanoparticles for ceftriaxone detection, it shows a typical sharp-band f-f transitions from Eu3+ responding to the antibiotics ceftriaxone, I recommend it to be published on Nanomaterials after minor revision, some questions and comments are listed below:

  1. 3, in the absorption spectrum, what’s the reason of absence of f-f transitions of Eu3+, and in the PL spectrum, the fine structure of the nano-particles is clearly observed, which often the case for bulk materials with well crystallization while for nanoparticles, inhomogeneous broadening is often observed, please make some comments.

The f-f transitions due to their low efficacy (they are forbidden) can be visualized in the spectra of lanthanide complexes in solid state, while only intraligand and/or ligand-metal charge transitions are observed in the electronic absorption spectra of europium complexes in solutions.

The main reason for inhomogeneous broadening of PL spectrum of nanoparticulate is a phase or size inhomogeneity of nanoparticles. The fine structures of the PL spectra of the nanoparticles agree well with the TEM and PXRD data, which reveal size and phase uniformity of the nanoparticles.

  1. 4, the lifetime of the sample is obviously over the detecting range of 10 ms, which equation did the author use for the fitting? Besides, the baseline should be carefully evaluated before lifetime fitting.

The lifetime was fitted by using a single exponential decay, accounting also for the baseline as a parameter of the fitting. The measured lifetime, as reported in the manuscript, was always below 10 ms. In particular, for the complexes 2 and 3 about 4 ms and for their PSS colloids about 3.5 ms. Anyway, we agree with the Referee that Figure 4 could be misleading, therefore it was corrected with the proper baseline.

Reviewer 2 Report

This work by Rustem Zairov et al. described “Phosphine oxide-Chelated Europium(III) Nanoparticles for Ceftriaxone Detection”. The paper is complete and well structured. The description of experimental procedure is detailed and results critically analyzed and commented. The crystallography is excellent. The data is fine.

I think the manuscript can be accepted in this form and it needs no revision.

Author Response

The authors thank the reviewer for the high evaluation of the work.

Reviewer 3 Report

The manuscript by Zairov and coworkers reports the synthesis of trivalent Eu and Tb complexes using geminal bisoxophosphoranes or CMPOs as neutral charged or anionic ligands. The solid state structures are elucidated via SCXRD for selected examples and the luminescence properties have been explored in solution. Complexes are then embedded in PSS stabilized colloids which are characterized with DLS and zeta potential and further explored for sensoric applications toward an antibiotic on preliminary basis.

There are some points requiring clarification or correction. At the end of section 3.4 the authors conclude from PXRD patterns that “the nanoparticles contain a pure crystalline phase of the complex without an admixture of the separately precipitated ligand phase”. However, they can only exclude admixture of additional crystalline ligand precipitate, whereas amorphous ligand or byproducts might well be admixed.

The sensoric tests with ceftriaxone as analyte species are only partially convincing. What are the relevant interactions between ceftriaxone and the metal ion? This could be explored for the complexes and the analyte without the colloid. Currently structurally quite diverse competing analytes have been tested, which totally lack the o- quinone site as prominent binding site in ceftriaxone. How about cross-response of the sensor towards other quinones or just simple oxalates?

IR data are reported for the ligands but not their complexes. The P=O vibrations and their shift upon coordination and colloid formation would be important to discuss as well since it can be determined in all phase regimes.

The introduction misses some relevant other work, for instance regarding CMPOs, Inorg. Chem. 2002, 41 (4) 727 and literature cited herein and regarding geminal bisPOs, Molecules 2023, 28(1), 48 and Eur. J. Inorg. Chem. 2022, 17, e202200194.

The abbreviation PSS is very common but still it needs to be defined upon first usage. Similarly, the captions of tables and figures are sometimes not very informative or incomplete (e.g. table 2; figure 4). Please clearly state whether data are for solid or solution phase.

Typos:

l 359: respetively

l 106/107: ESI or SI? please keep it consistent (BTW: caption of ESI/SI states “Electronic Supplementary material”

Figure 1: X should be CH not C

SI page 2: ppp

Formatted text in refs 5,26, 32

Author Response

The manuscript by Zairov and coworkers reports the synthesis of trivalent Eu and Tb complexes using geminal bisoxophosphoranes or CMPOs as neutral charged or anionic ligands. The solid state structures are elucidated via SCXRD for selected examples and the luminescence properties have been explored in solution. Complexes are then embedded in PSS stabilized colloids which are characterized with DLS and zeta potential and further explored for sensoric applications toward an antibiotic on preliminary basis.

There are some points requiring clarification or correction. At the end of section 3.4 the authors conclude from PXRD patterns that “the nanoparticles contain a pure crystalline phase of the complex without an admixture of the separately precipitated ligand phase”. However, they can only exclude admixture of additional crystalline ligand precipitate, whereas amorphous ligand or byproducts might well be admixed.

The reviewer is right that the PXRD pattern cannot exclude an admixture of amorphous byproducts, which are commonly manifested by the so-called amorphous galo. However, the PXRD pattern of the colloids indicates the insignificant contribution of amorphous phase, which is noted in the MS by the following sentences: “The PXRD pattern of the colloids reveals their crystalline nature with very poor if any contribution of amorphous phase (Figure 5 c, d). The PXRD pattern simulated out of single crystal data of [Eu(a)3] is also shown in Figure 5c for the comparison with that of PSS-[Eu(a)3] colloids in order to prove their composition.”

The insignificant levels of byproducts and “free” ligand agree well with the high stability of Eu(a)3. However, the enhanced activity of surface exposed complexes can trigger their partial dissociation. This phenomenon provides the specific activity of the surface exposed complexes, which contribute to the substrate-sensitivity of the nanomaterial.

The sensoric tests with ceftriaxone as analyte species are only partially convincing. What are the relevant interactions between ceftriaxone and the metal ion? This could be explored for the complexes and the analyte without the colloid. Currently structurally quite diverse competing analytes have been tested, which totally lack the o- quinone site as prominent binding site in ceftriaxone. How about cross-response of the sensor towards other quinones or just simple oxalates?

Thank you for raising the issue. In order to detail the reason for the sensing the following sentences have been included. “In turn, the dark complex formation observed for PSS-[Eu(a)3] colloids in the presence of ceftriaxone argues for the ligand exchange disturbing the ligand-to-metal energy transfer of [Eu(a)3]. The presence of the carboxylate and hydroxy group of 1,2,4-triazin-3-yl moiety in the structure of sodium ceftriaxone (Scheme S3, inset) prerequisites its efficiency in the ligand exchange.”

Unfortunately, the ceftriaxone-induced transformations of [Eu(a)3] in acetonitrile solutions, where the complex exists in molecular form, cannot model those in aqueous PSS-[Eu(a)3] colloids. This derives from the dipolar aprotic nature of acetonitrile and high proton donor ability of water molecules. Thus, the release of the deprotonated ligand a is thermodynamically favored in aqueous solutions, while this process is unfavorable in acetonitrile. Ceftriaxone is applied in the form of its sodium salt, which is well soluble in water, but its solubility in acetonitrile is low. Thus, both lower solubility and poorer dissociation of sodium ceftriaxone in acetonitrile vs aqueous solutions restrict studying of the ceftriaxone-induced transformations of [Eu(a)3] in acetonitrile solutions.

Commonly, a level of ceftriaxone is evaluated in blood serum or urine. For further use of PSS-[Eu(a)3] nanoparticles for ceftriaxone detection in blood serum, BSA, glutamic, and aspartic acids in 0.01 M phosphate buffer interfering effect was estimated. The luminescence response of PSS-[Eu(a)3] to ceftriaxone evaluated in the solutions modeling blood serum (Figure S4 in ESI) indicate that the applied concentration levels of the protein and aminoacids provide the insignificant effect on the sensitivity of PSS-[Eu(a)3] to ceftriaxone. This argues for an applicability of PSS-[Eu(a)3] for bioanalytical purposes.

IR data are reported for the ligands but not their complexes. The P=O vibrations and their shift upon coordination and colloid formation would be important to discuss as well since it can be determined in all phase regimes.

We thank author for suggestion. IR spectroscopy is widely used in structural studies of organic ligands as well as their complexes in solid state or when dissolved in organic solvents. In particular, IR spectra are widely applied to characterize complexes with C=O and P=O groups e.g. for Ligand b [Dalton Trans., 2017, 46, 15458–15469 doi: 10.1039/c7dt02678a] and Ligand a close structure analogue [Chem. Eur. J. 2002, 8, 24 5761–5771. doi:10.1002/1521-3765(20021216)8:24<5761::aid-chem5761>3.0.co;2-h] complexes IR data has been exhaustively represented in literature and we used it solely to confirm the structure. However, this study was mostly carried out in aqueous media hardly applicable for IR study. Complex of ligand d was out of the question since we did not continue the work with this derivative due to its poor characteristics showed at the very beginning. As to ligand c, IR data might be fruitful to describe complexes of this ligand but due to symmetrical structure and equivalence of the phosphine oxide moieties there is no doubts, which groups participate in coordination of the lanthanide also confirmed by XRD data. However, the single crystal XRD analysis is more powerful tool in analysis of the coordination mode of the complexes in the solid state than IR spectroscopy. Thus, the coordination mode of the complexes in crystal state is accurately and reliably estimated by the single crystal XRD analysis. The insignificant deviations between the PXRD pattern of the colloids and the pattern simulated out of single crystal data of [Eu(a)3] indicate the similarity of the complex structure in the single crystal samples and in the nanocrystallites of the colloidal samples.

Taking into account the impact of the ligand-to-metal energy transfer on the Eu3+-centered luminescence, its monitoring is the powerful tool to reveal the dissociation of the complex in diluted aqueous solutions. The similar information in the diluted aqueous solutions can be hardly obtained by the IR spectroscopy even in the case of the use of the specific techniques and high concentration (which is not always achievable). Thus, the applied in the present work methods are adequate to the nature of the subject and the surrounding medium.

The introduction misses some relevant other work, for instance regarding CMPOs, Inorg. Chem. 2002, 41 (4) 727 and literature cited herein and regarding geminal bisPOs, Molecules 202328(1), 48 and Eur. J. Inorg. Chem202217, e202200194.

 Thank you, these works have been included as references 25, 26, and 34.

The abbreviation PSS is very common but still it needs to be defined upon first usage. Similarly, the captions of tables and figures are sometimes not very informative or incomplete (e.g. table 2; figure 4). Please clearly state whether data are for solid or solution phase.

Definition of PSS abbreviation is now given at earliest use in the Experimental section. Captions were modified accordingly.

Typos:

l 359: respetively

l 106/107: ESI or SI? please keep it consistent (BTW: caption of ESI/SI states “Electronic Supplementary material”

Figure 1: X should be CH not C

SI page 2: ppp

Formatted text in refs 5,26, 32

All typos have been revised. “Electronic Supplementary material” was changed for “Electronic Supplementary Information”. ESI acronym was chosen when refer “Electronic Supplementary Information”. Titles of cited articles 5 and 32 (became 35) were also corrected in revised version of the MS. Reference 26 (shifted to 28 in updated version) is ok. That DOI is indeed that long and unusual.

Author Response

The authors reported series of lanthanide-centered luminous complexes and post-modified lanthanide-based nanoparticles. The sensing effect of the post-modified lanthanide-based nanoparticles on antibiotics was explored. The workload of this work seems to be a lot, but some key data are still not fully provided. Besides, some of the same letters appearing in the captions are confusing and not clear enough.

Q1: In the Figure 2d, please mark what element each colored atom represents for clarity.

Thanks for this remark. Each colored atom was marked with corresponding element symbols in the Figure 2d.

Q2: In the Figure 3-5, “1, 2, 3” in each figure and “a, b, c” in the caption are confusing and not clear enough. Please modify it further.

In the course of the entire manuscript, we chose the designation of ligands in bold letters (L=a, b, c). The regular a, b, c in parentheses ((a), (b), (c)) correspond to the panels of the figures. For clarity, the Arabic numerals in the Figure panels have been painted in the colors of the curves they relate to.

Q3: Please provide the relevant emission spectra of PSS-Eu for the recognition of amoxicillin and ciprofloxacin, or other similar antibiotics.

The spectra of PSS-[Eu(a)3] colloids at various concentrations of amoxicillin and ciprofloxacin were added as Figure S2 in ESI.

Q4: Whether PSS-Eu is reversible to the recognition of ceftriaxone? Please provide the results of the cycle experiment.

The luminescent response of the colloids to the analyte (ceftriaxone) derives from the quenching of Eu3+-centered luminescence under the binding of the analyte with PSS-[Eu(a)3] colloids. The high coordination capacity of the analyte argues for the ligand exchange disturbing the ligand-to-metal energy transfer of [Eu(a)3] as the reason for the quenching. Thus, the interaction of the analyte with PSS-[Eu(a)3] colloids is followed by the partial losses of the sensor after its interaction with the analyte. However, the easy synthetic procedure allows to convert [Eu(a)3] into PSS-[Eu(a)3] within half of an hour. Thus, the easy synthesis is good alternative to multiple washing steps required for recycling of sensing by another nanosensors, such as inorganic nanoparticles.

Q5: Please explain the mechanism of fluorescence quenching.

In order to detail the reason for the quenching the following sentences have been included. “In turn, the dark complex formation observed for PSS-[Eu(a)3] colloids in the presence of ceftriaxone argues for the ligand exchange disturbing the ligand-to-metal energy transfer of [Eu(a)3]. The presence of the carboxylate and hydroxy group of 1,2,4-triazin-3-yl moiety in the structure of disodium ceftriaxone (Scheme S3, inset) prerequisites its efficiency in the ligand exchange.”

Round 2

Reviewer 3 Report

All corrections have been performed and all questions have been appropriately answered

Reviewer 4 Report

I agree to accept the manuscript.